# CRISPR/Cas9-Mediated Gene Replacement in the Fungal Keratitis Pathogen *Fusarium solani var. petroliphilum*

**DOI:** 10.3390/microorganisms7100457

**Published:** 2019-10-16

**Authors:** Jorge D. Lightfoot, Kevin K. Fuller

**Affiliations:** 1Department of Microbiology and Immunology, University of Oklahoma Health Sciences Center, Oklahoma City, OK 73104, USA; 2Department of Ophthalmology, University of Oklahoma Health Sciences Center, Oklahoma City, OK 73104, USA

**Keywords:** *Fusarium solani*, *Fusarium solani* species complex, *Fusarium petroliphilum*, CRISPR/Cas9, genome editing, *ura3* gene, 5-FOA

## Abstract

Fungal keratitis (FK) is a site-threatening infection of the cornea associated with ocular trauma and contact lens wear. Members of the *Fusarium solani* species complex (FSSC) are predominant agents of FK worldwide, but genes that support their corneal virulence are poorly understood. As a means to bolster genetic analysis in FSSC pathogens, we sought to employ a CRISPR/Cas9 system in an FK isolate identified as *Fusarium petroliphilum*. Briefly, this approach involves the introduction of two components into fungal protoplasts: (1) A purified Cas9 protein complexed with guide RNAs that will direct the ribonuclease to cut on either side of the gene of interest, and (2) a “repair template” comprised of a hygromycin resistance cassette flanked by 40 bp of homology outside of the Cas9 cuts. In this way, Cas9-induced double strand breaks should potentiate double homologous replacement of the repair template at the desired locus. We targeted a putative *ura3* ortholog since its deletion would result in an easily discernable uracil auxotrophy. Indeed, 10% of hygromycin-resistant transformants displayed the auxotrophic phenotype, all of which harbored the expected *ura3* gene deletion. By contrast, none of the transformants from the repair template control (i.e., no Cas9) displayed the auxotrophic phenotype, indicating that Cas9 cutting was indeed required to promote homologous integration. Taken together, these data demonstrate that the in vitro Cas9 system is an easy and efficient approach for reverse genetics in FSSC organisms, including clinical isolates, which should enhance virulence research in these important but understudied ocular pathogens.

## 1. Introduction

The transparency of the cornea—due to its avascularity, paucity of resident cells, and careful assembly of collagen fibrils—allows for the entry of light into the eye for proper vision [1]. Following damage to the corneal epithelium, however, microbes can gain access to the underlying stroma and induce an inflammatory response (keratitis) that results in pain, photophobia and acute vision loss. Various microbes can cause keratitis (e.g., bacteria, amoeba and viruses), but fungi are associated with the highest rates of complications and poorest visual outcomes [2]. Approximately 20–40% of fungal keratitis (FK) cases do not resolve with antifungal intervention, resulting in the need for one or more corneal transplants [3].

Classical risk factors for the development of FK include tropical climate and ocular trauma [4]. In south India, for example, up to half of all microbial keratitis cases are fungal and such infections are common among agricultural laborers who develop corneal abrasions from vegetative debris [5]. A fungal etiology is rarer in the temperate United States (just 1.2% of keratitis cases are in New York), but becomes more common in the sub-tropical regions (16–24% in south Florida) [6,7]. Between 2005–2006, however, the Centers for Disease Control reported >250 cases of fungal keratitis in the U.S. and Asia associated with the Bausch and Lomb’s ReNU with MoistureLoc contact lens solution [8]. This outbreak challenged epidemiologic norms in that it occurred in patient groups (contact lens wearers) and regions (the temperate U.S.) previously thought to be at low risk for fungal keratitis. Approximately 34% of patients in the outbreak required corneal transplantation, thus underscoring the severity of these infections even in areas in which antifungals are readily available [8]. 

*Fusarium* species have emerged as predominant agents of FK worldwide, accounting for more than half of all fungal corneal ulcers in south India and every case in the above-mentioned Renu outbreak [4,8]. This is a large and diverse genus, however, consisting of least twenty monophyletic clades (called “species complexes”) that are distinguishable by molecular genotyping [9,10]. Sequencing of the internal transcribed spacer (ITS), situated between the 18S and 5.8S rRNA loci, can identify isolates at the species complex-level and has revealed that members of the *Fusarium solani* Species Complex (FSSC) account for 75–85% of *Fusarium* keratitis cases overall [11]. In a recent analysis of *Fusarium* keratitis cases in Florida, FSSC infections were also found to require longer treatment course, have worse follow-up visual acuity (BCVA), and have higher rates of corneal transplantation than non-FSSC infections [12]. These data therefore suggest that FSSC members are both the most prevalent and most virulent agents of *Fusarium* keratitis [13]. 

Sequence variation at the *translation elongation factor-1* (*tef-1*) locus can be used to place FSSC members into distinct molecular clades, some of which have been assigned separate species names. FSSC-clade 1, for example, is now called *F. petroliphilum* (sometimes *F. solani* var. *petroliphilum*), FSSC-2 is *F. keratoplaticum*, and FSSC-3 + 4 is *F. falciforme* [14,15,16]. The prevalence of these clades in the context of FK varies based on geography and etiology [17]. In south India, for example, *F. falciforme* accounts for ~80% of all FSSC keratitis cases [18]. This organism is also the most common *Fusarium* species isolated from soil and vegetation, which fits with FK resulting from agricultural-related trauma [19]. By contrast, *F. keratoplasticum* and *F. petroliphilum* are among the primary agents of *Fusarium* keratitis in non-tropical climates and were predominant in the Moisture Loc outbreak [14,20]. These species are common isolates in plumbing systems, which may correlate with their association with contact lens infections via tap water contamination [10,21]. 

Despite the clear importance of FSSC pathogens in the context of FK, little is known about the genes involved in their corneal virulence. The most powerful research approach toward this end is that of reverse genetics, in which candidate genes are targeted (e.g., deleted) followed by virulence testing of the mutant in an FK model. The lack of FSSC research of this sort likely reflects two related issues. First is an intrinsically low targeting efficiency due to a non-homologous end joining (NHEJ) machinery that promotes ectopic integration of the DNA construct (i.e., high false-positive ratio) [22]. The second is a lack of robust transformation protocols that, when coupled with the first issue, necessitates the screening of perhaps hundreds of colonies obtained from multiple, laborious transformation attempts. These problems have been overcome in other fungi by a myriad of CRISPR-based methodologies, but none have so far been developed in FSSC organisms. This disparity is the focus of our report. 

Clustered, regularly interspaced, short palindromic repeat (CRISPR) technologies require two critical components: 1) A CRISPR-associated (Cas9) endonuclease and, 2) a synthetic guide RNA (gRNA)—itself a fusion between CRISPR RNA (crRNA) and trans-activating crRNA (tracrRNA)—that associates with and directs Cas9 to a 20 nucleotide genomic target [23]. When this genomic target is directly next to protospacer-adjacent motif (PAM), typically an –NGG– sequence, Cas9 will induce a double-stranded break that can be repaired in one of several ways [24]. In the simplest iteration of the technique, the NHEJ machinery repairs the break within the gene’s open reading frame, but in doing so introduces an indel mutation that results in a frameshift [25]. The Cas9 cut can also be repaired by homologous recombination machinery when a “repair template” is introduced along with the CRISPR components [24]. In this way, the repair template may be a mutated or tagged allele that replaces the wild-type gene in situ. A third method, termed microhomology-mediated end joining (MMEJ), involves the introduction of short homology arms (~40 bp) on either side of a selectable marker (repair template) for directed gene replacement [26,27]. 

In the following work, we employ an in vitro assembled Cas9 ribonucleoprotein (RNP) complex coupled with a MMEJ repair template for targeted gene replacement in an FSSC keratitis isolate [27]. We target a putative *ura3* ortholog, encoding orotidine-5′-phosphate decarboxylase, as a case study and demonstrate that 10% of transformants display the expected phenotype and gene replacement event. We further demonstrate that the use of CRISPR increases the transformation yield overall, providing more colonies that can be screened and increasing the likelihood of isolating the desired mutant. Taken together, this methodology represents a simple and efficient means for targeted gene replacement in a clinical FSSC isolate, which should bolster reverse genetic studies in these important but understudied pathogens.

## 2. Materials and Methods

### 2.1. Fungal Isolates and Culture Media

All experiments were conducted with a clinical FK isolate (06-0110) from UCSF, previously identified as FSSC based on ITS sequencing [28]. Species-level identification was carried out by amplifying the *tef-1* locus with universal primers [29]. The PCR amplicon was subject to Sanger sequencing and the sequence was used in a BLASTn search against the Fusarium-ID database (isolate.fusariumdb.org). These results yielded a top hit for FSSC-1 (*Fusarium petroliphilum*).

Strains were cultured on glucose minimal medium (GMM) (1% glucose, clutterbuck salts, hunters trace elements, pH 6.5) or yeast, peptone, and dextrose medium (YPD; 2% dextrose, 2% peptone, 1% yeast extract) at 30 °C. 5 mM of uracil and uridine were added when specified. 2 g/L of 5-FOA was used in specified transformation selection media, and 1 g/L was used for further phenotyping [30]. Hygromycin B was used at a concentration of 200 µg/mL when necessary. Conidia were harvested and washed using PBS.

### 2.2. Design of crRNA’s for *ura3* in *F. solani* var *petroliphilum*

The amino acid sequence for PyrG in *Aspergillus nidulans* was used as a query in a BLASTp against the *Nectria haematococca* v2.0 Necha2_best_proteins database. It should be noted that *Nectria haematococca* is the teleomorphic name for *Fusarium solani*. This search returned one high homology hit with an E value of 3.3 × 10^−53^, Necha2199952, henceforth referred to as Ura3 the genomic sequence along with 500 bp flanking sequences were downloaded and used for further analysis and the design of crRNAs and primers.

### 2.3. Construction and Amplification of the Hygromycin Resistance Cassette

A hygromycin B phosphotransferase expression cassette was used as the selectable marker for the Cas9-mediated gene deletion throughout this work. A 2100 bp product spanning the *gpdA* promoter, the hygromycin B phosphotransferase gene, and the *trpC* terminator was amplified using Phusion polymerase (New England Biolabs, USA) from the plasmid pAN7 [31] using primers 1 and 2 (Table 1). The PCR fragments were purified using the E.Z.N.A Cycle Pure Kit and eluted in nuclease free water. The purified products were used as the microhomology mediated repair templates and contained a hygromycin B resistance cassette (*hygR*) flanked by 40 bp sequences homologous to the up and downstream regions of the targeted *ura3* gene. All primers used in this study are referenced in Table 1.

### 2.4. Cas9-gRNA Ribonucleoprotein Complexes

In Vitro assemblies were done as previously described in Al Abdallah et al. Briefly, Cas9 RNPs are composed of crRNA, tracrRNA, and the Alt-R® S.p. Cas9 Nuclease V3 (IDT 1081058) [27]. The gRNAs were assembled in vitro using 100 µM stock solutions of the lyophilized crRNA (Table 1) and the tracrRNA were prepared in nuclease free duplex buffer (Integrated DNA Technologies, USA) and combined in equimolar amounts, resulting in final concentrations of 33.3 µM concentrations of each [27]. This solution was heated to 95 °C for 5 min and allowed to cool to room temperature (20 °C) for 20 min. The resulting gRNAs were stored on ice for the short term and at −20 °C long-term. 

In order to form the Cas9 RNP 1.5 µL of each gRNA were separately combined with 0.75 µg of Cas9 in 11 µL of nuclease free Cas9 working buffer (20 mM HEPES, 150 mM KCl, pH 7.5). This mixture was incubated at 37 °C for 5 min to allow each separate Cas9 RNP to form before they were mixed (bringing the final volume to ~26.5 µL) and added to the *F. petroliphilum* protoplasts.

### 2.5. PEG-Mediated Fungal Transformation

Transformations were performed as previously described by Yelton et al. and Al Abdallah et al. with some modification [27,32]. Approximately 1 × 10^8^ microconidia were inoculated into YPD broth and germinated at 28 °C at 160 rpm overnight. The germlings were collected in miracloth and resuspended in 1.2 M KCl. The germlings were treated with a mixture of 5 mg/mL of Lysing Enzymes from *Trichoderma harzianum* (SIGMA), 5 mg/mL of Driselase (SIGMA, USA) and 100 µg/mL of Chitinase from *Streptomyces griseus* (SIGMA, USA). The germling and enzyme mixture was incubated at 28 °C for 2 h at 50 rpm. This suspension was monitored every 30 min microscopically for the formation of protoplasts.

Protoplasts were collected after 2 h incubation by centrifugation (5 min at 4000 *g* at 4 °C) and washed twice in STC50 buffer (1.2 M Sorbitol, 10 mM CaCl_2_, 10 mM Tris HCl pH 7.5). The protoplasts were resuspended in 1 mL of STC50 and enumerated on a hemocytometer, and diluted to a concentration of 2 × 10^7^. 200 µL of the protoplasts were used for each transformation. 

26.5 µL of the Cas9 RNPs and 10 µg of the repair template were combined with 200 µL of protoplasts and 25 µL of a 60% PEG solution (60% (*w*/*v*) PEG MW 3350, 50 mM CaCl_2_, 50 mM Tris HCl pH 7.5) and gently mixed by pipetting. This solution was placed on ice for 50 min. 1.25 mL of the 60% PEG solution was added and mixed by gently swirling and incubated at room temperature for 20 min. 1 mL of STC50 was added to each transformation and 150 µL were plated onto GMM, 1.2 M Sorbitol, 5 mM of Uracil and Uridine, 1.5% Agar, and 2 g/L of 5-FOA when indicated. After an overnight incubation 200 µg/mL of hygromycin B was introduced in an overlay (GMM with 0.5% Agar and 5mM Uracil and Uridine). Colonies were picked from the primary transformation plate onto minimal media supplemented with hygromycin, uracil, and uridine over the next 5–10 days of incubation at 30 °C. All genotyping PCRs, ed using NEB Hot Start *Taq* 2x Mastermix (New England Biolabs, USA) using the primers listed in Table 1 and the Figure 3 legend.

## 3. Results

### 3.1. Design of the crRNA Protospacer Sequences for Replacement of the *ura3* Gene with a Hygromycin Resistance Cassette

Our overall goal was to test the feasibility of using a CRISPR/Cas9 system for targeted gene replacement in a corneal isolate from the U.S., identified initially as *Fusarium* by culture and then placed within the FSSC by ITS sequencing [28]. For this study, we further identified the isolate as FSSC 1 (*F. petroliphilum*) by performing a BLASTn of the *tef-1* locus against the *Fusarium*-ID database. Accordingly, we will now refer to this organism as *F. petroliphilum*.

As a case study, we chose to target the *ura3* gene that, in other fungi, encodes the orotodine-5′-phosphate decarboxylase enzyme essential for uracil biosynthesis (Figure 1) [33]. The enzyme also converts exogenously added 5-fluoroorotic acid (5-FOA) to a toxic metabolite 5′-fluorouracil [34]. Accordingly, the successful replacement of the *ura3* gene with the *hygR* cassette would result in three easily assayable phenotypes: 1) Hygromycin resistance, 2) uracil auxotrophy, and 3) 5′-FOA resistance. To begin, we first identified the *ura3* ortholog in *F. petroliphilum* through a BLASTp search of the *Nectria haemotococca* v2.0 database (Mycocosm, Joint Genome Institute) using the *A. nidulans* ortholog, PyrG, as a query [35,36]. This yielded a single hit with 54.7% sequence identity (Necha2199952), henceforth referred to as *ura3*. 

We sought to employ an in vitro CRISPR–MMEJ method in which two components are introduced into fungal protoplasts [26,27,37]. First is the Cas9 ribonucleoprotein (RNP) complex, which consists of a commercially available Cas9 protein coupled with a gRNA. With this approach, the gRNA is assembled in vitro from two components: An invariable tracrRNA (also purchased commercially) and a target-specific CRISPR RNA (crRNA). Two distinct crRNAs were designed against genomic sequences (protospacers) within 100 bp of the putative *ura3* start and stop codons (UTRs). These protospacers (20 bp) not only harbored a PAM site (–NGG) at one end, as is essential, but they also contained one or more PAM sites within them as this has been suggested to increase Cas9 cutting efficiency and specificity (Figure 1C) [38]. Potential protospacer sequences were used as a query for a BLASTn search against the genome database to determine the possibility of any off-target cutting. In this way, protospacers that displayed more than 15 bp homology with a sequence preceding an off-target PAM motif were not used [39]. The second component required for the Cas9–MMEJ protocol is the repair template. Microhomology (40 bp) flanks, designed adjacent to the PAM site and crRNA protospacers, were incorporated into the *hygR* cassette by PCR. The design of both the microhomology flanks and the crRNA protospacers are illustrated in Figure 1C and the primer sequences are listed in Table 1. 

In short, two Cas9 RNPs were generated in vitro such that cutting would occur in *ura3* 5′ and 3′ UTRs. These cuts would, in principle, promote the replacement of the *ura3* coding sequence with the repair template by homologous recombination via the microhomology flanks.

### 3.2. Cas9–MMEJ Transformation Results in Several Colonies of the Expected ura3 Deletion Phenotype

To analyze the efficacy of the above-described CRISPR strategy, three *F. petroliphilum* protoplast transformation groups were analyzed: (1) Control, which received no Cas9 RNPs or repair template DNA; (2) repair template only, which received 10 µg of repair template DNA; and (3) Cas9–MMEJ, which received both 10 µg of the repair template and the two flanking Cas9 RNP complexes. Protoplasts from each group were initially recovered on, and then subcultured onto, minimal media supplemented with hygromycin, uracil and uridine. In this way, stable transformants would grow regardless of whether the repair template integrated homologously or ectopically. As reflected in Table 2, the total number of hygromycin resistant colonies was ~5 times higher in the Cas9–MMEJ group compared to the repair template only group, suggesting that the Cas9 RNPs increased the transformation efficiency in some way. As expected, the control group yielded no hygromycin-resistant colonies.

To determine if any of the hygromycin-resistant transformants were also uracil auxotrophs, the colonies were subcultured onto minimal media supplemented with either (1) uracil/uridine, (2) hygromycin and uracil/uridine, (3) neither. In addition to the transformants, the wild-type strain was included on each medium as a control. Figure 2 shows a subset of the Cas9–MMEJ transformants screened in this way, all of which were able to grow in the presence of hygromycin as expected. Six colonies of those screened (~10%) were unable to grow in the absence of uracil/uridine, indicating that they were *ura3* deletants (auxotrophs). By contrast, none of the “repair template only” transformants were uracil auxotrophs, suggesting that they all had the *hygR* cassette present as an ectopic integration (Table 2, and data not shown).

In addition to being auxotrophic for uracil/uridine, *ura3* deletion mutants should also be resistant to 5-FOA. Indeed, whereas the recipient wild-type strain was completely growth inhibited on 5-FOA-containing medium, the uracil/uridine auxotrophs grew otherwise normally after a brief delay (Figure 2B). Taken together, the phenotypic data suggests that a proportion of the hygromycin-resistant transformants recovered from the Cas9–MMEJ transformation were the desired *ura3* replacements.

### 3.3. Genotyping of the Uracil Auxotrophic Mutants Confirms the Expected *ura3* Gene Replacement Event

To characterize the genotype of the uracil auxotrophs, we first performed a PCR of the *ura3* locus with primers (primers ¾, Table 1) situated outside of the expected area of recombination(Figure 3A). The repair template is approximately 100 bp larger than the *ura3* coding sequence and, as expected, this band shift was apparent in all of the auxotrophic mutants analyzed. We then used the *ura3* locus amplicon (Figure 3A) as a template for another PCR with nested primers (primers 7/8, Table 1), for the *hygR* cassette. All auxotrophic mutants yielded a band in this assay, indicating that the *ura3* coding sequence was replaced with the *hygR* cassette (Figure 3B). To confirm that the *ura3* coding sequence was indeed absent from the genome in these mutants, we next performed a PCR on the genomic DNA with primers situated within the coding region (primers 5/6, Table 1). Again, as expected, the wild-type strain yielded a predicted amplicon whereas the auxotrophic mutants did not (Figure 3C). 

Cumulatively, these results indicate that the auxotrophic phenotypes observed in our Cas9–MMEJ experiment were due to the expected recombination event.

## 4. Discussion

The early CRISPR systems developed in fungi involved the integration of *cas9* and single guide (sg)RNA cassettes in the organism’s genome [40]. In the absence of a repair template, gene disruption is dependent upon the error-prone NHEJ machinery to induce nonsense mutations or otherwise non-functional gene products [41]. By contrast, the inclusion of MHEJ repair templates allowed for a greater versatility of applications, including site-directed mutagenesis and in situ gene tagging [42]. The presence of such repair templates also increases the overall targeting efficiency of the system. The Cas9-mediated disruption of *pksP* in *A. fumigatus*, for example, increased from approximately 45% by NHEJ to 95–100% with the inclusion of a MMEJ repair template [43]. In any case, such methodologies raise an important concern of whether the presence of the *cas9* gene within the cell would have long-term, genotoxic effects. Indeed, it is estimated that gRNAs, on average, can have up to five mismatch (off-target) genes, and the constitutive presence of *cas9* likely increases the probability they will be mutated [44,45]. Although Fuller et al. demonstrated that constitutive expression of *cas9* did not impact growth or virulence of *A. fumigatus*, Foster et al. recently reported that *cas9* expression was highly toxic to *Magnaporthe oryzae* [37,46]. This suggests that Cas9 can mediate unintended and deleterious mutations, at least in some fungal species. In attempt to mitigate such effects, conditional promoter systems have been developed, such as *pNiiA* in *A. fumigatus*, as well as “suicide system” in *Cryptococcus neoformans* in which the *cas9* cassette is excised from the genome after editing [43,47]. 

The in vitro-assembled Cas9 approach, in principle, largely bypasses the concern of long-term *cas9* toxicity and off-target effects [46,48]. As this system involves the transient introduction of purified Cas9 RNPs into protoplasts, there is no genetic material that can be integrated into the genome beyond the repair template. Indeed, Abdallah et al. performed whole-genome sequencing of *A. fumigatus* mutants generated by this approach and found that Cas9 RNPs did not induce more secondary mutations than traditional homologous recombination techniques [48]. The activity of Cas9 may vary between species, however, and so our future studies will employ whole-genome sequencing in this and other CRISPR-mediated *F. petroliphilum* mutants to determine the cryptic mutation rate. 

A further advantage of the in vitro systems is their ease of use. Each of the in vitro-assembled RNP components may be obtained commercially, so the only molecular biology required ahead of transformation is PCR amplification of the selectable marker (repair template) with primers that incorporate the MMEJ flanks. The resulting PCR amplicon can then be used directly in the transformation, thus negating the need for any cloning step. 

In this study, an in vitro-assembled Cas9–MMEJ system was used to successfully replace the *ura3* ortholog in a *F. petroliphilum* corneal isolate. Based on screening hygromycin-resistant transformants for uracil auxotrophy followed by genotyping, we describe a gene targeting efficiency of 10%. This efficiency is relatively low compared to some fungal wild-type backgrounds in which a similar CRISPR strategy was employed, including *A. fumigatus* (50–75%), *Magnaporthe oryzae* (70–80%), and *C. neoformans* (81–87%) [27,46,47]. However, our rates are comparable with other fungal species, including *Fusarium oxysporum* (21%) and *Trichoderma reesei* (3–30%), suggesting that there is a strong species-dependent influence on targeting efficiency [26,49]. The presence of a robust NHEJ machinery likely accounts for ectopic integration of the repair template and this can be overcome by utilizing backgrounds deficient in essential NHEJ genes. For example, the targeting of *pksP* in *A. fumigatus* increased from 50–75% in a wild-type strain to 95–100% in an ∆*akuB* (ku80) mutant [27]. However, the power of CRISPR, at least for our interests, lies within the ability to transform clinical isolates. We noticed that the inclusion of Cas9 RNPs increased the number of hygromycin-resistance colonies overall five-fold, from 16 in the “template only” group to 83. Therefore, a 10% targeting efficiency makes it probable that several mutants of interest will be isolated within a single transformation attempt. Future efforts will focus on optimizing the protocol by adjusting the MMEJ homology length and/or Cas9 RNP concentrations. 

In summary, we report the first use, to our knowledge, of CRISPR/Cas9 technology for reverse genetics in *F. petroliphilum* or any FSSC species. An appreciation of the FSSC in the context of keratitis and systemic fusariosis has increased in recent years, but an understanding of genes that govern their pathogenesis has not. With improved genetic systems in place, including the in vitro Cas9–MMEJ system described here, we hope to bolster virulence-related research in these important pathogens and improve treatment modalities in the patients affected by them.

## Figures and Tables

**Figure 1 microorganisms-07-00457-f001:**
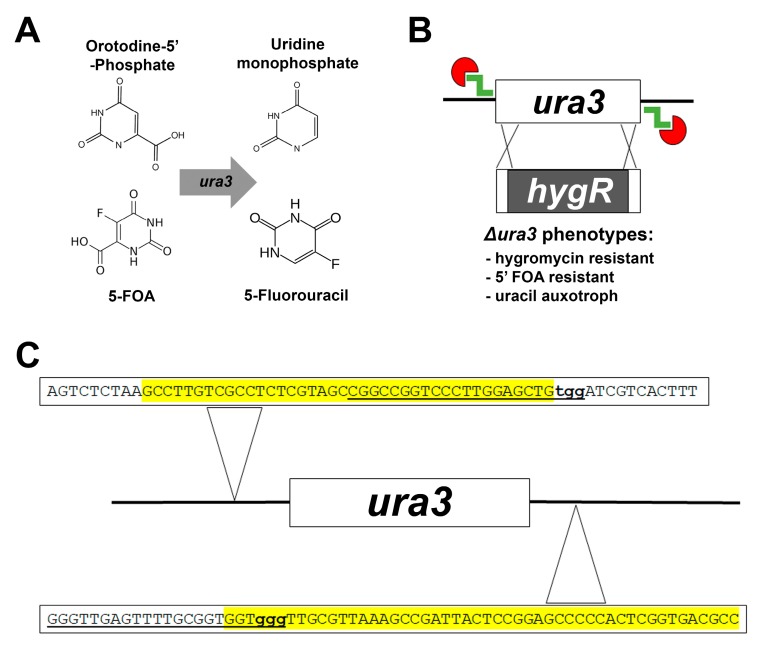
An overview of the *ura3* deletion strategy using in vitro assembled Cas9 RNPs coupled with microhomology-mediated end joining (MMEJ). (**A**) *Ura3* decarboxylates orotodine-5′-monophosphate to form uridine monophosphate, an essential uracil precursor, as well as 5-fluorotic acid (FOA), added exogenously, to form a toxic metabolite 5-fluorouracil. (**B**) Cas9 ribonucleoprotein (RNP)’s target the up and downstream regions of the *ura3* gene and the MMEJ repair replaces *ura3* with *hygR* resulting in uracil auxotrophy, 5-FOA resistance, and hygromycin resistance. (**C**) The regions of microhomology for the repair template are highlighted in yellow, the crRNA sequences are underlined, and the protospacer-adjacent motif (PAM) sites are bolded in lowercase.

**Figure 2 microorganisms-07-00457-f002:**
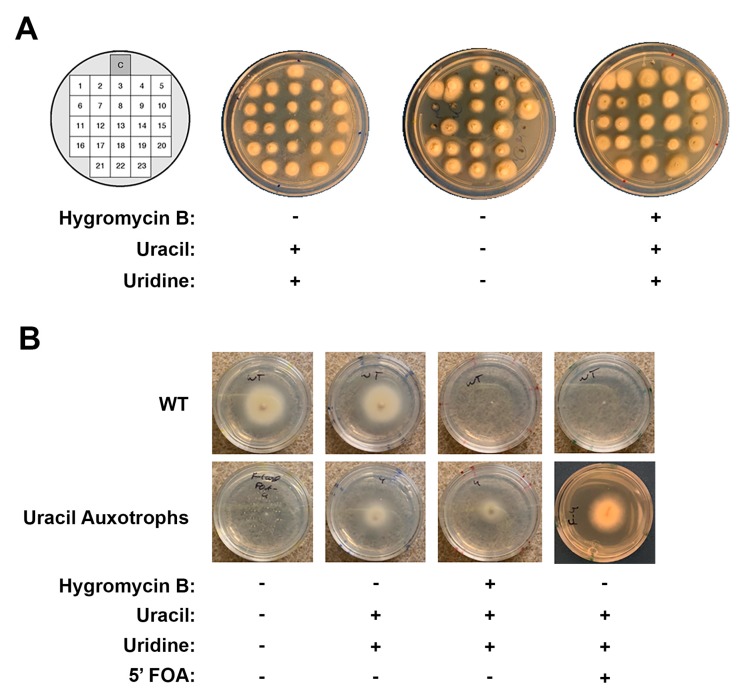
Phenotyping transformants for uracil auxotrophy and 5-FOA resistance. (**A**) Transformants were picked and replicated across three growth conditions with each position on the plate corresponding to a single transformant. The topmost position is the wild-type control. Transformants in positions 6, 7, 11, 15 and 20 displayed the desired uracil auxotrophy phenotype (center plate). (**B**) A subculture of an uracil auxotroph as compared to the wild type control. Incubation times varied from 5–10 days at 30 °C. Incubation on Media containing 5-FOA needed 10 days for comparable growth to media without 5-FOA.

**Figure 3 microorganisms-07-00457-f003:**
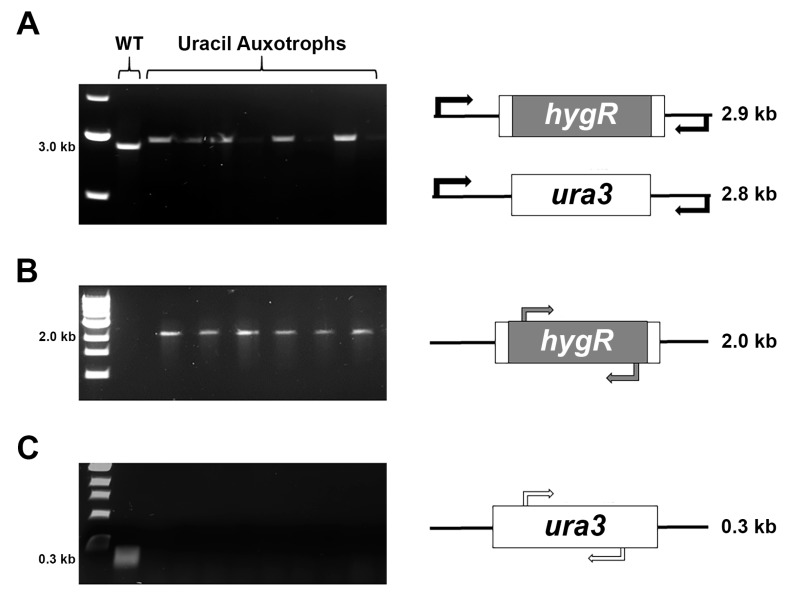
Genotyping of uracil auxotrophs. (**A**) Amplification of the *ura3* locus using primers 3 and 4 yields the expected 100 bp band shift between the wild-type and the uracil auxotrophs. (**B**) Amplification of the *hygR* cassette from the amplicon in Figure 3A using primers 7 and 8. The uracil auxotrophs contain the hygromycin resistance cassette while the wild-type strain does not. (**C**) Amplification of the *ura3* coding region from genomic DNA using primers 5 and 6. Only the wild-type contains the coding sequence for *ura3*.

**Table 1 microorganisms-07-00457-t001:** Oligonucleotides and crRNA’s used for this study.

Amplification of Hygromycin Resistance Cassette with Ura3 Microhomology
Primer 1:
5′ GCCTTGTCGCCTCTCGTAGCCGGCCGGTCCCTTGGAGCTGaagtggaaaggctggtgtgc
Primer 2:
5′ AGTGGGGGCTCCGGAGTAATCGGCTTTAACGCAACCCACCtcgcgtggagccaagagcgg
**Genotyping putative *ura3* deletants**
Primer 3: 5′ CAAGCCAAGCTTCGCACAAG
Primer 4: 5′ GCGATGACATTCAGTGCAGC
Primer 5: 5′ GTCGAGCTGCAATACACCAG
Primer 6: 5′ GACAAGACGTGGTGAATCGG
Primer 7: 5′ AAGTGGAAAGGCTGGTGTGC
Primer 8: 5′ TCGCGTGGAGCCAAGAGCGG
**crRNA sequences**
5′ crRNA FsUra3: 5′ CGGCCGGTCCCTTGGAGCTG
3′ crRNA FsUra3: 5′ GGGTTGAGTTTTGCGGTGGT

**Table 2 microorganisms-07-00457-t002:** Results of the PEG mediated transformation. A subset of hygromycin resistant colonies were subcultured onto various media. Uracil auxotrophs could not grow on media that did not contain uracil.

	1° Plate (Hyg Selection)	Analyzed	Hyg Resistant	Uracil Auxotrophs	Efficiency
Control	0	0	0	0	0%
Repair Template only	16	10	10	0	0%
Cas9–MMEJ	83	63	63	6	9.50%

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
