# Peer review of "CRISPR/Cas9-Mediated Gene Replacement in the Fungal Keratitis Pathogen Fusarium solani var. petroliphilum"

_microorganisms, 2019, doi:10.3390/microorganisms7100457_

Round 1

Reviewer 1 Report

This article describes the creation of a Cas9 RNP complex coupled with a MMEJ repair template for replacement of a target gene, in this case the ura3 gene. The design is very simple and effective and is clearly described by the authors. As always off target effects are of concern with CRISPR but the authors state that as Cas9 expression is transient it is unlikely and refer to an article in which whole genome sequencing was performed in another species using this method as evidence that they would not get any off target effects in their transformants. The authors have not performed whole genome sequencing in their transformants so this can only be inferred not concluded. Is it possible to perform whole genome sequencing of any of these transformants?  

Overall I think this paper is clearly written, easy to follow, has enough detail for someone to reproduce it and is of interest to fungi researchers. I have some minor corrections or comments as listed below.

Minor corrections:

Figure 1A needs to have better resolution some parts are very hard to read

In the methods section it should be mentioned that Necteria haematococca is also known as Fusarium solani for those not in this field.

In Figure 3A the WT presumably stands for wild type but I think it is above the ladder rather than the wt lane. This and the gels in Fig3B and 3C would also benefit from some size markers

L40 change due to do

L60 remove the

L72 add of after importance

L92 change are to or

L99 remove that (one of them)

L131 space after BLAStp

L133 was to were

Consistency in methods with a space between the number and the unit

L248 change the to then

L268 change cassete to cassette

L297 change out to our

Reviewer 2 Report

The manuscript by Lightfoot and Fuller submitted for its consideration to Microorganisms journal, describes the use of gene editing to selectively modify the genome of a strain of Fusarium solani involved in fungal keratitis. The protocol described is sound, interesting and very clever and will open new possibilities for the characterization of the pathogenicity determinants in this fungal strain. Protocols are well detailed, which could also help other scientists within the field to use them in other fungal pathogenic strains of Fusarium.

I sincerely think that the manuscript is within the scope of the journal and very interesting for the readers. I would recommend it for publication, after the authors clarify some very minor points.

Point-to-point comments

1.- Please, normalize the species denomination along the text. The authors used F. petroliphilium to name the F. solani var, but it should be completed to include the original genus.

2.- A technical question arises from the text. The authors described as a proof of concept the targeted deletion of ura3 gene as an example of how the system could be used. How difficult is to expand the universe of Fusarium genetic variants?. Are they so easy to target by the CRISPR system?. Did the authors perform some tests to edit other pathogenic strains from clinical isolates?
